# On Optimal Robustness to Adversarial Corruption in Online Decision Problems

**Shinji Ito**
NEC Corporation
i-shinji@nec.com

## Abstract

This paper considers two fundamental sequential decision-making problems: the problem of prediction with expert advice and the multi-armed bandit problem. We focus on stochastic regimes in which an adversary may corrupt losses, and we investigate what level of robustness can be achieved against adversarial corruption. The main contribution of this paper is to show that optimal robustness can be expressed by a square-root dependency on the amount of corruption. More precisely, we show that two classes of algorithms, anytime Hedge with decreasing learning rate and algorithms with second-order regret bounds, achieve $O(\frac{\log N}{\Delta} + \sqrt{\frac{C \log N}{\Delta}})$-regret, where $N, \Delta$, and $C$ represent the number of experts, the gap parameter, and the corruption level, respectively. We further provide a matching lower bound, which means that this regret bound is tight up to a constant factor. For the multi-armed bandit problem, we also provide a nearly-tight lower bound up to a logarithmic factor.

## 1 Introduction

In this work, we consider two fundamental sequential decision-making problems, the problem of prediction with expert advice (expert problem) and the multi-armed bandit (MAB) problem. In both, a player chooses probability vectors $p_t$ over a given action set $[N] = \{1, 2, \ldots, N\}$ in a sequential manner. More precisely, in each round $t$, a player chooses a probability vector $p_t \in [0, 1]^N$ over the action set, and then an environment chooses a loss vector $\ell_t \in [0, 1]^N$. The player chooses $p_t$, and then observes $\ell_t$ in the expert problem. In the MAB problem, the player picks action $i_t \in [N]$ following $p_t$ and then observes $\ell_{ti_t}$. The goal of the player is to minimize the (pseudo-) regret $\bar{R}_T$ defined as

$$R_{Ti^*} = \sum_{t=1}^{T} \ell_t^\top p_t - \sum_{t=1}^{T} \ell_{ti^*}, \quad \bar{R}_{Ti^*} = \mathbf{E}\left[R_{Ti^*}\right], \quad \bar{R}_T = \max_{i^* \in [N]} \bar{R}_{Ti^*}. \tag{1}$$

For such decision-making problems, two main types of environments have been studied: stochastic environments and adversarial environments. In stochastic environments, the loss vectors are assumed to follow an unknown distribution, i.i.d. for all rounds. It is known that the difficulty of the problems can be characterized by the *suboptimality gap* parameter $\Delta > 0$, which denotes the minimum gap between the expected losses for the optimal action and for suboptimal actions. Given the parameter $\Delta$, mini-max optimal regret bounds can be expressed as $\Theta(\frac{\log N}{\Delta})$ in the expert problem [Degenne and Perchet, 2016, Mourtada and Gaïffas, 2019] and $\Theta(\frac{N \log T}{\Delta})$ in the MAB problem [Auer et al., 2002a, Lai and Robbins, 1985, Lai, 1987]. In contrast to the stochastic model, the adversarial model does not assume any generative models for loss vectors, but the loss at each round may behave adversarially depending on the choices of the player up until that round. The mini-max optimal regret bounds for the adversarial model are $\Theta(\sqrt{T \log N})$ in the expert problem [Cesa-Bianchi and Lugosi, 2006,

35th Conference on Neural Information Processing Systems (NeurIPS 2021).

Table 1: Regret bounds in stochastic regimes with adversarial corruption

| Problem setting | Upper bound | Lower bound |
|---|---|---|
| Expert problem | $O\left(\frac{\log N}{\Delta} + C\right)$[1] [Amir et al., 2020] | $\Omega\left(\frac{\log N}{\Delta} + \sqrt{\frac{C \log N}{\Delta}}\right)$ |
| | $O\left(\frac{\log N}{\Delta} + \sqrt{\frac{C \log N}{\Delta}}\right)$ [Theorems 3, 4] | [Theorem 5] |
| Multi-armed bandit | $O\left(\frac{N \log T}{\Delta} + \sqrt{\frac{CN \log T}{\Delta}}\right)$ [Zimmert and Seldin, 2021] | $\Omega\left(\frac{N}{\Delta} + \sqrt{\frac{CN}{\Delta}}\right)$ [Theorem 6] |

Freund and Schapire, 1997] and $\Theta(\sqrt{TN})$ in the MAB problem [Audibert and Bubeck, 2009, Auer et al., 2002b].

As can be seen in the regret bounds, achievable performance differs greatly between the stochastic and adversarial regimes, which implies that the choice of models and algorithms will matter in many practical applications. One promising solution to this challenge is to develop *best-of-both-worlds (BOBW)* algorithms, which perform (nearly) optimally in both stochastic and adversarial regimes. For the expert problem, Gaillard et al. [2014] provide an algorithm with a BOBW property, and Mourtada and Gaïffas [2019] have shown that the well-known Hedge algorithm with decreasing learning rate (decreasing Hedge) enjoys a BOBW property as well. For the MAB problem, the Tsallis-INF algorithm by Zimmert and Seldin [2021] has a BOBW property, i.e., achieves $O(\frac{N \log T}{\Delta})$-regret in the stochastic regime and $O(\sqrt{NT})$-regret in the adversarial regime. One limitation of BOBW guarantees is, however, that they do not necessarily provide nontrivial regret bounds for a situation in which the stochastic and the adversarial regimes are *mixed*, i.e., an intermediate situation.

To overcome this BOBW-property limitation, our work focuses on an intermediate (and comprehensive) regime between the stochastic and adversarial settings. More specifically, we consider the *adversarial regime with a self-bounding constraint* introduced by Zimmert and Seldin [2021]. As shown by them, this regime includes the *stochastic regime with adversarial corruption* [Lykouris et al., 2018, Amir et al., 2020] as a special case in which an adversary modifies the i.i.d. losses to the extent that the total amount of changes does not exceed $C$, which is an unknown parameter referred to as the *corruption level*. For expert problems in the stochastic regime with adversarial corruption, Amir et al. [2020] have shown that the decreasing Hedge algorithm achieves $O(\frac{\log N}{\Delta} + C)$-regret. For the MAB problem, Zimmert and Seldin [2021] have shown that Tsallis-INF achieves $O(\frac{N \log T}{\Delta} + \sqrt{\frac{CN \log T}{\Delta}})$-regret in adversarial regimes with self-bounding constraints. To be more precise, they have proved a regret upper bound of $O(\sum_{i \neq i^*} \frac{\log T}{\Delta_i} + \sqrt{C \sum_{i \neq i^*} \frac{\log T}{\Delta_i}})$, where $i^* \in [N]$ is the optimal action and $\Delta_i$ represents the suboptimality gap for each action $i \in [N] \setminus \{i^*\}$. Ito [2021b] has shown that similar regret bounds hold even when there are multiple optimal actions, i.e., even if the number of actions $i$ with $\Delta_i = 0$ is greater than 1. In such cases, the terms of $\sum_{i \neq i^*} 1/\Delta_i$ in regret bounds are replaced with $\sum_{i:\Delta_i > 0} \frac{1}{\Delta_i}$. In addition to this, Masoudian and Seldin [2021] have improved the analysis to obtain a refined regret bound of $O\left(\left(\sum_{i \neq i^*} \frac{1}{\Delta_i}\right) \log_+\left(\frac{(N-1)T}{(\sum_{i \neq i^*} \frac{1}{\Delta_i})^2}\right) + \sqrt{C\left(\sum_{i \neq i^*} \frac{1}{\Delta_i}\right) \log_+\left(\frac{(N-1)T}{C \sum_{i \neq i^*} \frac{1}{\Delta_i}}\right)}\right)$, where $\log_+(x) = \max\{1, \log x\}$.

The contributions of this work are summarized in Table 1, alongside previously reported results. As shown in Theorems 3 and 4, this paper provides an improved regret upper bound of $O(\frac{\log N}{\Delta} + \sqrt{\frac{C \log N}{\Delta}})$ for the expert problem in the adversarial regime with self-bounding constraints. This regret upper bound is tight up to a constant factor. In fact, we provide a matching lower bound in Theorem 5. In addition to this, we show an $\Omega(\frac{N}{\Delta} + \sqrt{\frac{CN}{\Delta}})$-lower bound for MAB, which implies that

---

[1]Note that Amir et al. [2020] adopt a different definition of regret than in this paper. Details and notes for comparison are discussed in Remark 1.

Tsallis-INF by Zimmert and Seldin [2021] achieves a nearly optimal regret bound up to an $O(\log T)$ factor in the adversarial regime with self-bounding constraints.

The regret bounds in Theorems 3 and 4 are smaller than the regret bound shown by Amir et al. [2020] for the stochastic regime with adversarial corruption, especially when $C = \Omega(\frac{\log N}{\Delta})$, and they can be applied to more general problem settings in the adversarial regime with self-bounding constraints. Note here that this study and their study consider slightly different definitions of regret. More precisely, they define regret using losses *without* corruption, while this study uses losses *after* corruption to define regret. In practice, appropriate definitions would vary depending on individual situations. For example, if each expert's prediction is itself corrupted, the latter definition is suitable. In contrast, if only the observation of the player is corrupted, the former definition seems appropriate. However, even after taking this difference in definitions into account, we can see that the regret bound in our work is, in a sense, stronger than theirs, as is discussed in Remark 1 of this paper. In particular, we would like to emphasize that the new bound of $O(\frac{\log N}{\Delta} + \sqrt{\frac{C \log N}{\Delta}})$ provides the first theoretical evidence implying that the corresponding algorithms are more robust against adversarial corruption than the naive Follow-the-Leader algorithm. Note also that the regret bound by Amir et al. [2020] is tight as long as the former regret definition is used.

This work shows the tight regret upper bounds for two types of known algorithms. The first, (Theorem 3), is the decreasing Hedge algorithm, which has been analyzed by Amir et al. [2020] and Mourtada and Gaïffas [2019] as well. The second (Theorem 4) represents algorithms with *second-order regret bounds* [Cesa-Bianchi et al., 2007, Gaillard et al., 2014, Hazan and Kale, 2010, Steinhardt and Liang, 2014, Luo and Schapire, 2015]. It is worth mentioning that Gaillard et al. [2014] have shown that a kind of second-order regret bounds implies $O(\frac{\log N}{\Delta})$-regret in the stochastic regime. Theorem 4 in this work extends their analysis to a broader setting of the adversarial regime with self-bounding constraints. In the proof of Theorems 3 and 4, we follow a proof technique given by Zimmert and Seldin [2021] to exploit self-bounding constraints.

To show regret lower bounds in Theorems 5 and 6, we construct specific environments with corruption that provide insight into effective attacks which would make learning fail. Our approach to corruption is to modify the losses so that the optimality gaps decrease. This approach achieves a mini-max lower bound in the expert problem (Theorem 5) and a nearly-tight lower bound in MAB up to a logarithmic factor in $T$ (Theorem 6). We conjecture that there is room for improvement in this lower bound for MAB under assumptions regarding consistent policies [Lai and Robbins, 1985], and that the upper bound by Zimmert and Seldin [2021] is tight up to a constant factor.

## 2   Related work

In the context of the expert problem, studies on stochastic settings seem to be more limited than those on adversarial settings. De Rooij et al. [2014] have focused on the fact that the Follow-the-Leader (FTL) algorithm works well for a stochastic setting, and they have provided an algorithm that combines FTL and Hedge algorithms to achieve the best of both worlds. Gaillard et al. [2014] have provided an algorithm with a second-order regret bound depending on $V_{Ti^*} = \sum_{t=1}^{T}(\ell_t^\top p_t - \ell_{ti^*})^2$ in place of $T$ and have shown that such an algorithm achieves $O(\frac{\log N}{\Delta})$-regret in the stochastic regime. Mourtada and Gaïffas [2019] have shown that a simpler Hedge algorithm with decreasing learning rates of $\eta_t = \Theta(\sqrt{\frac{\log N}{t}})$ enjoys a tight regret bound in the stochastic regime as well. This simple decreasing Hedge algorithm has been shown by Amir et al. [2020] to achieve $O(\frac{\log N}{\Delta} + C)$-regret in the stochastic regime with adversarial corruption. For online linear optimization, a generalization of the expert problem, Huang et al. [2016] have shown that FTL achieves smaller regret in the stochastic setting and provides best-of-both-worlds algorithms via techniques reported by Sani et al. [2014].

For MAB, there are a number of studies on best-of-both-worlds algorithms [Bubeck and Slivkins, 2012, Zimmert and Seldin, 2021, Seldin and Slivkins, 2014, Seldin and Lugosi, 2017, Pogodin and Lattimore, 2020, Auer and Chiang, 2016, Wei and Luo, 2018, Zimmert et al., 2019, Lee et al., 2021, Ito, 2021b]. Among these, studies by Wei and Luo [2018], Zimmert and Seldin [2021], Zimmert et al. [2019] are closely related to this work. In their studies, gap-dependent regret bounds in the stochastic regime are derived via $\{p_t\}$-dependent regret bounds in the adversarial regime, similarly to that seen in this work and in previous studies by Gaillard et al. [2014], Amir et al. [2020].

Studies on online optimization algorithms robust against adversarial corruption have been extended to a variety of models, including those for the multi-armed bandit [Lykouris et al., 2018, Gupta et al., 2019, Zimmert and Seldin, 2021, Hajiesmaili et al., 2020], Gaussian process bandits [Bogunovic et al., 2020], Markov decision processes [Lykouris et al., 2021, Chen et al., 2021, Jin et al., 2021], the problem of prediction with expert advice [Amir et al., 2020], online linear optimization [Li et al., 2019], and linear bandits [Bogunovic et al., 2021, Lee et al., 2021]. Literature can also be found regarding effective attacks on bandit algorithms [Jun et al., 2018, Liu and Shroff, 2019].

As summarized by Hajiesmaili et al. [2020], there can be found studies on two different models of adversarial corruption: the oblivious corruption model and the targeted corruption model. In the former (e.g., in studies by Lykouris et al. [2018], Gupta et al. [2019], Bogunovic et al. [2020]), the attacker may corrupt the losses $\ell_t$ after observing $(\ell_t, p_t)$ without knowing the chosen action $i_t$ while, in the latter (e.g., in studies by Jun et al. [2018], Hajiesmaili et al. [2020], Liu and Shroff [2019], Bogunovic et al. [2021], Ito [2021a], Erez and Koren [2021]), the attacker can choose corruption depending on $(\ell_t, p_t, i_t)$. We discuss differences between these models in Section 3. This work mainly focuses on the oblivious corruption model for MAB problems. It is worth mentioning that Tsallis-INF [Zimmert and Seldin, 2021] works well in the oblivious corruption models, as is shown in Table 1, as well as achieving best-of-both-worlds.

## 3 Problem setting

A player is given $N$, the number of actions. In each round $t = 1, 2, \ldots$ the player chooses a probability vector $p_t = (p_{t1}, p_{t2}, \ldots, p_{tN})^\top \in \{p \in [0,1]^N \mid \|p\|_1 = 1\}$, and then the environment chooses a loss vector $\ell_t = (\ell_{t1}, \ell_{t2}, \ldots, \ell_{tN})^\top \in [0,1]^N$. In the expert problem, the player can observe all entries of $\ell_t$ after outputting $p_t$. By way of contrast, in the MAB problem, the player picks $i_t$ w.r.t. $p_t$, i.e., chooses $i_t$ so that $\mathrm{Prob}[i_t = i|p_t] = p_{ti}$, and then observes $\ell_{ti_t}$. The performance of the player is measured by means of the regret defined in (1).

Note that in MAB problems we have

$$\mathbf{E}\left[\sum_{t=1}^{T}\left(\ell_{ti_t} - \ell_{ti^*}\right)\right] = \mathbf{E}\left[\sum_{t=1}^{T}\left(\ell_t^\top p_t - \ell_{ti^*}\right)\right] = \bar{R}_{Ti^*} \tag{2}$$

under the assumption that $\ell_t$ is independent of $i_t$, given $p_t$.

This paper focuses on environments in the following regime:

**Definition 1** (Adversarial regime with a self-bounding constraint [Zimmert and Seldin, 2021]). We say that the environment is in an *adversarial regime with a* $(i^*, \Delta, C, T)$ *self-bounding constraint* if

$$\bar{R}_{Ti^*} = \mathbf{E}\left[\sum_{t=1}^{T}(\ell_t^\top p_t - \ell_{ti^*})\right] \geq \Delta \cdot \mathbf{E}\left[\sum_{t=1}^{T}(1 - p_{ti^*})\right] - C \tag{3}$$

holds for any algorithms, where $\Delta \in [0,1]$ and $C \geq 0$.

In this paper, we deal with the situation in which the player is *not* given parameters $(i^*, \Delta, C, T)$.

The regime defined in Definition 1 includes the following examples:

**Example 1** (Stochastic regime). Suppose $\ell_t \in [0,1]^N$ follows an unknown distribution $\mathcal{D}$ over i.i.d. for $t \in \{1, 2, \ldots\}$. Denote $\mu = \mathbf{E}_{\ell \sim \mathcal{D}}[\ell]$, and let $i^* \in \arg\min_{i \in [N]} \mu_i$ and $\Delta = \min_{i \in [N] \setminus \{i^*\}}(\mu_i - \mu_{i^*})$. The environment is then in the adversarial regime with a self-bounding constraint (3) with $C = 0$. Note here that $\Delta > 0$ implies that the optimal action is *unique*, i.e., $\mu_i > m_{i^*}$ holds for any action $i \in [N] \setminus \{i^*\}$.

**Example 2** (Adversarial regime). If we set $\Delta = 1$ and $C = 2T$, for any $i^* \in [N]$ and for any algorithms, arbitrary loss sequences $\{\ell_t\}_{t=1}^T \subseteq [0,1]^N$ satisfy (3). This means that the (purely) adversarial environment will be in the adversarial regime with a $(i^*, 1, 2T, T)$ self-bounding constraint for any $i^* \in [N]$ and $T$.

**Example 3** (Stochastic regime with adversarial corruption). Suppose $\ell_t \in [0,1]^N$ is given as follows: (i) a temporary loss $\ell_t' \in [0,1]^N$ is generated from an unknown distribution $\mathcal{D}$ (i.i.d. for $t$) and (ii) an adversary corrupts $\ell_t'$ after observing $p_t$ to determine $\ell_t$ subject to the constraint of

$\sum_{t=1}^{T} \| \mathbf{E}[\ell_t] - \mathbf{E}[\ell_t'] \|_\infty \le C$. As shown in [Zimmert and Seldin, 2021], this regime satisfies (3), i.e., is a special case of the adversarial regime with a self-bounding constraint.

**Remark 1.** For the stochastic regime with adversarial corruption, different notions of regret can be found in the literature. An alternative to the definition in (1) is regret w.r.t. losses without corruption, i.e., $R'_{Ti^*} = \sum_{t=1}^{T} \left( \ell_t'^{\top} p_t - \ell_{ti^*}' \right)$, ($\bar{R}'_{Ti^*}$, and $\bar{R}'_T$ can also be defined in a similar way). In general, which metric will be appropriate depends on the situation in which the algorithm is applied. For example, in the case of prediction with expert advice, if each expert's prediction is itself corrupted, the player's performance should be evaluated in terms of the regret $\bar{R}_T$, as the consequential prediction performance is determined by the losses $\ell_t$ after corruption, not by $\ell_t'$. In contrast, if only the observation of the player is corrupted, the performance should be evaluated in terms of $\bar{R}'_T$.

We can easily see that $|\bar{R}'_{Ti^*} - \bar{R}_{Ti^*}| \le 2C$. Amir et al. [2020] have shown a regret bound of $\bar{R}'_T = O\left(\frac{\log N}{\Delta} + C\right)$, which immediately implies $\bar{R}_T = O\left(\frac{\log N}{\Delta} + C\right)$. Similarly, a regret bound of $\bar{R}_T = O\left(\frac{\log N}{\Delta} + \sqrt{\frac{C \log N}{\Delta}}\right)$ immediately implies $\bar{R}'_T = O\left(\frac{\log N}{\Delta} + C\right)$. In fact, from the AM-DM inequality, we have

$$\frac{\log N}{\Delta} + \sqrt{\frac{C \log N}{\Delta}} \le \frac{\log N}{\Delta} + \frac{1}{2}\left(C + \frac{\log N}{\Delta}\right) = O\left(C + \frac{\log N}{\Delta}\right).$$

Note here that the former bound of $\bar{R}_T = O\left(\frac{\log N}{\Delta} + \sqrt{\frac{C \log N}{\Delta}}\right)$ is properly stronger than the latter of $\bar{R}'_T = O\left(\frac{\log N}{\Delta} + C\right)$, as the latter does not necessarily imply the former.

**Remark 2.** In MAB, a *targeted corruption model* has been considered to be a variant of the model in Example 3. In this model, the adversary corrupts the losses after observing $i_t$. In this case, the loss $\ell_t$ after corruption and $i_t$ are dependent given $p_t$, and hence (2) does not always hold.

## 4   Regret upper bound

### 4.1   Known regret bounds for adversarial regimes by hedge algorithms

The Hedge algorithm [Freund and Schapire, 1997] (also called the multiplicative weight update [Arora et al., 2012] or the weighted majority forecaster [Littlestone and Warmuth, 1994]) is known to be a mini-max optimal algorithm for the expert problem. In the Hedge algorithm, the probability vector $p_t$ is defined as follows:

$$w_{ti} = \exp\left(-\eta_t \sum_{j=1}^{t-1} \ell_{ji}\right), \quad p_t = \frac{w_t}{\|w_t\|_1}, \tag{4}$$

where $\eta_t > 0$ are learning rate parameters. If $p_t$ is given by (4), the regret is bounded as follows:

**Lemma 1.** *If $\{p_t\}_{t=1}^{T}$ is given by* (4) *with decreasing learning rates (i.e., $\eta_t \ge \eta_{t+1}$ for all $t$), for any $\{\ell_t\}_{t=1}^{T}$ and $i^* \in [N]$, the regret is bounded as*

$$R_{Ti^*} \le \frac{\log N}{\eta_1} + \sum_{t=1}^{T}\left(\frac{1}{\eta_t}\sum_{i=1}^{N} p_{ti} g\left(\eta_t(-\ell_{ti} + \alpha_t)\right) + \left(\frac{1}{\eta_{t+1}} - \frac{1}{\eta_t}\right) H(p_{t+1})\right), \tag{5}$$

*for any $\{\alpha_t\}_{t=1}^{T} \subseteq \mathbb{R}$, where $g$ and $H$ are defined as*

$$g(x) = \exp(x) - x - 1, \quad H(p) = \sum_{i=1}^{N} p_i \log \frac{1}{p_i}. \tag{6}$$

From this lemma, using $g(x) \approx x^2/2$ and $H(p) \le \log N$, we obtain the following regret bounds for adversarial settings:

**Theorem 1** (Theorem 2.3 in [Cesa-Bianchi and Lugosi, 2006]). *If $p_t$ is given by* (4) *with $\eta_t = \sqrt{\frac{8 \log N}{t}}$, for any $T$, $i^* \in [N]$ and $\{\ell_t\}_{t=1}^{T} \subseteq [0,1]^N$, the regret is bounded as*

$$R_{Ti^*} \le \sqrt{2T \log N} + \log N/8. \tag{7}$$

Hedge algorithms with decreasing learning rates $\eta_t = \Theta(\sqrt{\frac{\log N}{t}})$, as in Theorem 1, are referred to as decreasing Hedge, e.g., in [Mourtada and Gaïffas, 2019]. Such algorithms are shown by Mourtada and Gaïffas [2019] to achieve $O(\sqrt{\frac{\log N}{\Delta}})$-regret in stochastic regimes, and are also shown, by Amir et al. [2020], to achieve $O(\sqrt{\frac{\log N}{\Delta}} + C)$-regret in stochastic regimes with adversarial corruption.

Besides such worst-case regret bounds as found in Theorem 1, a variety of data-dependent regret bounds have also been developed (see, e.g., [Steinhardt and Liang, 2014]). One remarkable example is that of the *second-order* bounds by Cesa-Bianchi et al. [2007], which depend on parameters $V_T$ defined as follows:

$$v_t = \sum_{i=1}^{N} p_{ti}(\ell_{ti} - \ell_t^\top p_t)^2, \quad V_T = \sum_{t=1}^{T} v_t \tag{8}$$

A regret bound depending on $V_T$ rather than $T$ can be achieved wih the following adaptive learning rates:

**Theorem 2** (Theorem 5 in [Cesa-Bianchi et al., 2007])**.** *If $p_t$ is given by* (4) *with,* $\eta_t = \min\left\{1, \sqrt{\frac{2(\sqrt{2}-1)\log N}{(e-2)V_{t-1}}}\right\}$, *the regret is bounded as*

$$R_{Ti^*} \leq 4\sqrt{V_T \log N} + 2\log N + 1/2 \tag{9}$$

*for any $T$, $i^*$ and $\{\ell_t\}_{t=1}^T \subseteq [0,1]^N$.*

As $V_T \leq T$ follows from the definition (8), the regret bound in (2) includes the worst-case regret bound of $R_{Ti^*} = O(\sqrt{T \log N})$. Further, as shown in Corollary 3 of [Cesa-Bianchi et al., 2007], the bound in Theorem 2 implies $R_{Ti^*} = O(\sqrt{\frac{L_{Ti^*}(T-L_{Ti^*})}{T}\log N})$, where $L_{Ti^*} = \sum_{t=1}^T \ell_{ti^*}$. This means that the regret will be improved if the cumulative loss $L_{Ti^*}$ for optimal action $i^*$ is small or is close to $T$.

### 4.2 Refined regret bound for decreasing Hedge

This subsection shows that the algorithm described in Theorem 1 enjoys the following regret bound as well:

**Theorem 3.** *If $p_t$ is given by* (4) *with $\eta_t = \sqrt{\frac{8\log N}{t}}$, we have*

$$\bar{R}_{Ti^*} \leq 100\frac{\log N}{\Delta} + 10\sqrt{\frac{C\log N}{\Delta}} \tag{10}$$

*under the assumption that* (3) *holds.*

*Proof.* Using the fact that $g(x) \leq \frac{(e-1)x^2}{2}$ for $x \leq 1$ and $H(p) \leq (1-p_{i^*})(1+\log N - \log(1-p_{i^*}))$, from Lemma 1, we obtain

$$R_{Ti^*} \leq 9\log N + \frac{1}{4\sqrt{\log N}} \sum_{t=1}^{T} \frac{1-p_{ti^*}}{\sqrt{t}}\left(12\log N + \log\frac{1}{1-p_{ti^*}}\right). \tag{11}$$

A proof for (11) can be found in the appendix. From (3) and (11), for any $\lambda > 0$, we have

$$\bar{R}_{Ti^*} = (1+\lambda)\bar{R}_{Ti^*} - \lambda\bar{R}_{Ti^*}$$

$$\leq \mathbf{E}\left[(1+\lambda)\left(9\log N + \frac{1}{4\sqrt{\log N}}\sum_{t=1}^{T}\frac{1-p_{ti^*}}{\sqrt{t}}\left(12\log N + \log\frac{1}{1-p_{ti^*}}\right)\right) - \lambda\left(\Delta\sum_{t=1}^{T}(1-p_{ti^*}) - C\right)\right]$$

$$\leq 9(1+\lambda)\log N + \lambda C + \frac{1+\lambda}{4\sqrt{\log N}}\mathbf{E}\left[\sum_{t=1}^{T}\frac{1-p_{ti^*}}{\sqrt{t}}\left(12\log N - \frac{4\lambda\Delta\sqrt{t\log N}}{1+\lambda} - \log(1-p_{ti^*})\right)\right].$$

To bound the values of the expectation, we use the following inequality

$$\sum_{t=1}^{T} \frac{x_t}{\sqrt{t}} \left( a - b\sqrt{t} - \log x_t \right) \leq \frac{2a^2 + 1}{b} + b \tag{12}$$

which holds for any $a, b > 0$, $T$ and $\{x_t\}_{t=1}^{T} \subseteq (0, 1)$. A proof of (12) is given in the appendix. Combining the above two displayed inequalities with $a = 12 \log N$ and $b = \frac{4\lambda\Delta\sqrt{\log N}}{1+\lambda}$, we obtain

$$\bar{R}_{Ti^*} \leq 9(1+\lambda) \log N + \lambda C + \frac{1+\lambda}{4\sqrt{\log N}} \left( (2(12 \log N)^2 + 1)\frac{1+\lambda}{4\lambda\Delta\sqrt{\log N}} + \frac{4\lambda\Delta\sqrt{\log N}}{1+\lambda} \right)$$

$$= 9(1+\lambda) \log N + \lambda C + \frac{(1+\lambda)^2}{\lambda\Delta \log N} \cdot \left( 18(\log N)^2 + \frac{1}{16} \right) + \Delta$$

$$\leq 9 \log N + \Delta + \frac{38 \log N}{\Delta} + \lambda \left( 9 \log N + C + \frac{19 \log N}{\Delta} \right) + \frac{1}{\lambda}\frac{19 \log N}{\Delta}.$$

By choosing $\lambda = \sqrt{(\frac{19 \log N}{\Delta})/(9 \log N + C + \frac{19 \log N}{\Delta})}$, we obtain

$$\bar{R}_{Ti^*} \leq 9 \log N + \Delta + \frac{38 \log N}{\Delta} + 2\sqrt{\left( 9 \log N + C + \frac{19 \log N}{\Delta} \right)\frac{19 \log N}{\Delta}}$$

$$\leq 9 \log N + \Delta + \frac{38 \log N}{\Delta} + 2\sqrt{532 \left( \frac{\log N}{\Delta} \right)^2 + \frac{19C \log N}{\Delta}}$$

$$\leq 9 \log N + \Delta + (38 + 2\sqrt{532})\frac{\log N}{\Delta} + 2\sqrt{19}\sqrt{\frac{C \log N}{\Delta}} \leq 100\frac{\log N}{\Delta} + 10\sqrt{\frac{C \log N}{\Delta}}$$

where the third inequality follows from $\sqrt{x + y} \leq \sqrt{x} + \sqrt{y}$ for $x, y \geq 0$. □

Combining Theorems 1 and 3, we can see that the decreasing Hedge with $\eta_t = \sqrt{\frac{8 \log N}{t}}$ achieves $\bar{R}_{Ti^*} = O(\min\{\frac{\log N}{\Delta} + \sqrt{\frac{C \log N}{\Delta}}, \sqrt{T \log N}\})$ in the adversarial regive with self-bounding constraints. This bound will be shown to be tight up to a constant factor in Section 5.

### 4.3 Refined regret bound for adaptive Hedge

In this subsection, we show that a second-order regret bound as seen in Theorem 2 implies tight gap-dependent regret bounds in the adversarial regime with a self-bounding constraint.

We start from the observation that $v_t$ defined in (8) satisfies $v_t \leq (1 - p_{ti^*})$ for any $i^*$. In fact, we have $v_t \leq \sum_{i=1}^{N} p_{ti}(\ell_{ti} - \alpha)^2$ for any $\alpha \in \mathbb{R}$ as the right-hand side is minimized when $\alpha = \ell_t^\top p_t$, from which it follows that

$$v_t \leq \sum_{i=1}^{N} p_{ti}(\ell_{ti} - \ell_{ti^*})^2 = \sum_{i \in [N]\setminus\{i^*\}} p_{ti}(\ell_{ti} - \ell_{ti^*})^2 \leq \sum_{i \in [N]\setminus\{i^*\}} p_{ti} = 1 - p_{ti^*}. \tag{13}$$

Hence, the regret bound in Theorem 2 implies

$$R_{Ti^*} \leq 4\sqrt{\log N \sum_{t=1}^{T}(1 - p_{ti^*}) + 2 \log N} + \frac{1}{2}. \tag{14}$$

Such a regret bound depending on $\sum_{t=1}^{T}(1 - p_{ti^*})$ leads to a tight gap-dependent regret bound, as shown in the following theorem:

**Theorem 4.** *Suppose that the regret is bounded as*

$$R_{Ti^*} \leq \sqrt{A \sum_{t=1}^{T}(1 - p_{ti^*}) + B}. \tag{15}$$

*Then, under the condition of* (3)*, the pseudo-regret is bounded as*

$$\bar{R}_{Ti^*} \le \frac{A}{\Delta} + B + \sqrt{\frac{A(B+C)}{\Delta}}. \tag{16}$$

*Proof.* From (15) and (3), for any $\lambda > 0$ we have

$$\bar{R}_{Ti^*} = (1+\lambda)\bar{R}_{Ti^*} - \lambda\bar{R}_{Ti^*}$$

$$\le \mathbf{E}\left[ (1+\lambda)\left( \sqrt{A\sum_{t=1}^{T}(1-p_{ti^*})} + B \right) - \lambda\left( \Delta\sum_{t=1}^{T}(1-p_{ti^*}) - C \right) \right]$$

$$\le \frac{A(1+\lambda)^2}{4\lambda\Delta} + (1+\lambda)B + \lambda C = \frac{A}{2\Delta} + B + \frac{1}{\lambda}\frac{A}{4\Delta} + \lambda\left( \frac{A}{4\Delta} + B + C \right),$$

where the second inequality follows from $a\sqrt{x} - bx = -b(\sqrt{x} - \frac{a}{2b})^2 + \frac{a^2}{4b} \le \frac{a^2}{4b}$ for $a > 0$, $b \in \mathbb{R}$ and $x \ge 0$. By choosing $\lambda = \sqrt{(\frac{A}{4\Delta})/(\frac{1}{4\Delta} + B + C)}$ we obtain

$$\bar{R}_{Ti^*} \le \frac{A}{2\Delta} + B + 2\sqrt{\frac{A}{4\Delta}\cdot\left(\frac{A}{4\Delta} + B + C\right)} = \frac{A}{2\Delta} + B + \sqrt{\left(\frac{A}{2\Delta}\right)^2 + \frac{A(B+C)}{\Delta}}$$

$$\le \frac{A}{2\Delta} + B + \frac{A}{2\Delta} + \sqrt{\frac{A(B+C)}{\Delta}} = \frac{A}{\Delta} + B + \sqrt{\frac{A(B+C)}{\Delta}},$$

where the second inequality follows from $\sqrt{x+y} \le \sqrt{x} + \sqrt{y}$ for $x, y \ge 0$. $\square$

Combining this theorem and (14), we obtain the following regret bound for the algorithm described in Theorem 2:

**Corollary 1.** *If $p_t$ is chosen by* (4) *with $\eta_t = \min\{1, \sqrt{\frac{2(\sqrt{2}-1)\log N}{(e-2)V_{t-1}}}\}$, under the condition of* (3)*, the pseudo regret is bounded as $\bar{R}_{Ti^*} \le 16\frac{\log N}{\Delta} + 4\sqrt{\frac{(3\log N + C)\log N}{\Delta}} + 3\log N$.*

Theorem 4 can be applied to algorithms other than the one in Theorem 2. One example is an algorithm by Gaillard et al. [2014]. In Corollary 8 of their paper, a regret bound of $R_{Ti^*} \le C_1\sqrt{\log N \sum_{t=1}(\ell_t^\top p_t - \ell_{ti^*})^2} + C_2$ is provided. Then, as it holds that $(\ell_t^\top p_t - \ell_{ti^*})^2 \le 1 - p_{ti^*}$, we have (15) with appropriate $A$ and $B$, and, consequently, we obtain the regret bound given in (16).

## 5 Regret lower bound

This section provides (nearly) tight lower bounds for the expert problem and the MAB problem in the adversarial regime with a self-bounding constraint. Let us begin by describing the statement for the expert problem:

**Theorem 5.** *For any $\Delta \in (0, 1/4)$, $N \ge 4$, $T \ge 4\log N$, $C \ge 0$, and for any algorithm for the expert problem, there exists an environment in the adversarial regime with a $(i^*, \Delta, N, C, T)$ self-bounding constraint for which the pseudo-regret is at least*

$$\bar{R}_{Ti^*} = \Omega\left( \min\left\{ \frac{\log N}{\Delta} + \sqrt{\frac{C\log N}{\Delta}}, \sqrt{T\log N} \right\} \right). \tag{17}$$

To show this lower bound, we define a distribution $\mathcal{D}_{\Delta,i^*}$ over $\{0,1\}^N$ for $\Delta \in (0, 1/4)$ and $i^* \in [N]$, as follows: if $\ell \sim \mathcal{D}_{\Delta,i^*}$, $\ell_{i^*}$ follows a Bernoulli distribution of parameter $1/2 - \Delta$, i.e., $\text{Prob}[\ell_{i^*} = 1] = 1/2 - \Delta$ and $\text{Prob}[\ell_{i^*} = 0] = 1/2 + \Delta$, and $\ell_i$ follows a Bernoulli distribution of parameter $1/2$ for $i \in [N] \setminus i^*$, independently. We then can employ the following lemma:

**Lemma 2** (Proposition 2 in [Mourtada and Gaïffas, 2019]). *For any algorithm and for any $\Delta \in (0, 1/4)$, $N \ge 4$ and $T \ge \frac{\log N}{\Delta^2}$, there exists $i^*$ such that $\bar{R}_{Ti^*} \ge \frac{\log N}{256\Delta}$ holds for $(\ell_t)_{t=1}^T \sim \mathcal{D}_{\Delta,i^*}^T$.*

Using this lower bound, we can show Theorem 5.

*Proof of Theorem 5.* We show lower bounds for the following four cases: (i) If $T \leq \frac{\log N}{\Delta^2}$, $\bar{R}_{Ti^*} = \Omega(\sqrt{T \log N})$. (ii) If $\frac{C}{\Delta} \leq \frac{\log N}{\Delta^2} \leq T$, $\bar{R}_{Ti^*} = \Omega(\frac{\log N}{\Delta})$. (iii) If $\frac{\log N}{\Delta^2} \leq \frac{C}{\Delta} \leq T$, $\bar{R}_{Ti^*} = \Omega(\sqrt{\frac{C \log N}{\Delta}})$. (iv) If $\frac{\log N}{\Delta^2} \leq T \leq \frac{C}{\Delta}$, $\bar{R}_{Ti^*} = \Omega(\sqrt{T \log N})$. Combining all four cases of (i)–(iv), we obtain (17).

(i) Suppose $T < \frac{\log N}{\Delta^2}$. Set $\Delta' = \sqrt{\frac{\log N}{T}}$. We then have $T = \frac{\log N}{\Delta'^2}$ and $\Delta < \Delta' \leq 1/4$. If $\ell_t \sim \mathcal{D}_{\Delta',i^*}$ for all $t \in [T]$, the environment is then in an adversarial regime with a $(i^*, \Delta, N, C, T)$ self-bounding constraint for any $C \geq 0$, and the regret is bounded as $\bar{R}_{Ti^*} \geq \frac{\log N}{256\Delta'} = \Omega(\sqrt{T \log N})$ from Lemma 2.

(ii) Suppose $\frac{C}{\Delta} \leq \frac{\log N}{\Delta^2} \leq T$. If $\ell_t \sim \mathcal{D}_{\Delta,i^*}$ for all $t \in [T]$, the regret is bounded as $\bar{R}_{Ti^*} \geq \frac{\log N}{256\Delta}$ for some $i^*$ from Lemma 2. The environment is then in an adversarial regime with a $(i^*, \Delta, N, C, T)$ self-bounding for any $C \geq 0$.

(iii) Suppose $\frac{\log N}{\Delta^2} \leq \frac{C}{\Delta} \leq T$. Define $\Delta' = \sqrt{\frac{\Delta \log N}{C}} \leq \Delta$. We then have $\frac{\log N}{\Delta'} = \sqrt{C \frac{\log N}{\Delta}}$. Let $T' = \lceil \frac{\log N}{\Delta'^2} \rceil = \lceil \frac{C}{\Delta} \rceil \leq T$. Consider an environment in which $\ell_t \sim \mathcal{D}_{\Delta',i^*}$ for $t \in [T']$ and $\ell_t \sim \mathcal{D}_{\Delta,i^*}$ for $t \in [T'+1, T]$. Then from Lemma 2, there exists $i^* \in [N]$ such that $\bar{R}_{Ti^*} \geq \bar{R}_{T'i^*} \geq \frac{\log N}{\Delta'} = \Omega(\sqrt{\frac{C \log N}{\Delta}})$. Further, we can show that the environment is in an adversarial regime with a $(i^*, \Delta, N, C, T)$ self-bounding constraint. In fact, we have $T'(\Delta - \Delta') \leq \frac{C}{\Delta}(\Delta - \Delta') \leq C$.

(iv) Suppose $\frac{\log N}{\Delta^2} \leq T \leq \frac{C}{\Delta}$. Set $\Delta' = \sqrt{\frac{\log N}{T}}$ and consider $\ell_t \sim \mathcal{D}_{\Delta',i^*}$ for all $t \in [T]$. The regret is then bounded as $\bar{R}_{Ti^*} \geq \frac{\log N}{256\Delta'} = \Omega(\sqrt{T \log N})$ for some $i^*$, from Lemma 2. We can confirm that the environment is in an adversarial regime with a $(i^*, \Delta, N, C, T)$ self-bounding constraint, as we have $\Delta' T \leq \Delta T \leq C$, where the first and second inequalities follow from $\frac{\log N}{\Delta^2} \leq T$ and $T \leq \frac{C}{\Delta}$, respectively. □

Via a similar strategy to that used in this proof, we can show the regret lower bound for the MAB problem as well:

**Theorem 6.** *For any $\Delta \in (0, 1/4)$, $N \geq 4$, $T \geq 4 \log N$, $C \geq 0$, and for any multi-armed bandit algorithm, there exists an environment in the adversarial regime with a $(i^*, \Delta, N, C, T)$ self-bounding constraint for which the pseudo-regret is at least*

$$\bar{R}_{Ti^*} = \Omega\left(\min\left\{\frac{N}{\Delta} + \sqrt{\frac{CN}{\Delta}}, \sqrt{NT}\right\}\right). \tag{18}$$

We can demonstrate this theorem by means of the following lemma:

**Lemma 3** ([Auer et al., 2002b]). *For any multi-armed bandit algorithm and for any $\Delta \in (0, 1/4)$, $N \geq 4$ and $T \geq \frac{N}{\Delta^2}$, there exists $i^*$ such that $\bar{R}_{Ti^*} \geq \frac{N}{32\Delta}$ holds for $(\ell_t)_{t=1}^T \sim \mathcal{D}_{\Delta,i^*}^T$.*

A complete proof of Theorem 6 can be found in the appendix.

# 6 Discussion

In this paper, we have shown $O(R + \sqrt{CR})$-regret bounds for the expert problem, where $R$ stands for the regret bounds for the environment without corruption and $C$ stands for the corruption level. From the matching lower bound, we can see that this $O(\sqrt{CR})$-term characterizes optimal robustness against corruption.

One natural question is whether such $O(R + \sqrt{CR})$-type regret bounds can be found for other online decision problems, such as online linear optimization, online convex optimization, linear bandits, and convex bandits. To our knowledge, algorithms achieving $O(R + \sqrt{CR})$-regret are known at this time for the expert problem, for the MAB problem [Zimmert and Seldin, 2021], for the combinatorial semi-bandit problem [Zimmert et al., 2019, Ito, 2021a], and the problem of online learning with feedback graphs [Erez and Koren, 2021]. What these algorithms have in common is that they use a

framework of Follow-the-Regularized-Leader with decreasing learning rates and that they achieve the best-of-both-worlds simultaneously. As Amir et al. [2020] suggest, online mirror descent algorithms does not have $O(R + \sqrt{CR})$-regret bounds, in contrast to the Follow-the-Regularized-Leader case. We believe that characterizing algorithms with $O(R + \sqrt{CR})$-regret bounds will be important for future work.

Another question is whether we can remove the assumption that the optimal action is unique. This assumption is required for the analysis of most algorithms with $O(R + \sqrt{CT})$-regret bounds [Zimmert et al., 2019, Ito, 2021a, Erez and Koren, 2021]. One exception to this is the Tsallis-INF algorithm by Zimmert and Seldin [2021], which was shown by Ito [2021b] to not require the uniqueness assumption. It is not yet known, however, whether the technique by Ito [2021b] can be applied to the expert problem.

**On Potential Societal Impact**    This study focuses on the robustness of online optimization algorithms and includes a discussion of effective attacks on algorithms. The regret upper bounds obtained in this study suggest that the impact of the attacker on the learning outcome is rather limited. Hence, at this point, we do not see any particular negative social consequences. Researchers working on the theory of online optimization and adversarial robustness may benefit from this paper.

## Acknowledgments and Disclosure of Funding

The author was supported by JST, ACT-I, Grant Number JPMJPR18U5, Japan.

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
