# A Appendix

## A.1 Proof of Lemma 1

The Hedge algorithm defined in (4) can be interpreted as a special case of follow-the-regularized leader (FTRL) methods, as follows:

$$p_t \in \arg\min_{p \in [0,1]^N : \|p\|_1 = 1} \left\{ \sum_{j=1}^{t-1} \ell_j^\top p - \frac{1}{\eta_t} H(p) \right\}. \tag{19}$$

From a standard analysis of FTRL (see, e.g., Exercise 28.12 in the book by Lattimore and Szepesvári [2020], where we set $F_t(x) = -\frac{1}{\eta_t} H(x)$ ), we have

$$R_T \le \sum_{t=1}^{T} \left( \ell_t^\top (p_t - p_{t+1}) - \frac{1}{\eta_t} KL(p_{t+1}, p_t) \right) + \frac{H(p_1)}{\eta_1} + \sum_{t=1}^{T} \left( \frac{1}{\eta_{t+1}} - \frac{1}{\eta_t} \right) H(p_{t+1}), \tag{20}$$

where $KL(p, q)$ represents the KL divergence defined by $KL(p, q) = \sum_{i=1}^{N} (p_i \log \frac{p_i}{q_i} - p_i + q_i)$. For any $\alpha_t \in \mathbb{R}$, the first term in the right-hand side can be bounded as

$$\ell_t^\top (p_t - p_{t+1}) - \frac{1}{\eta_t} KL(p_{t+1}, p_t) = (\ell_t - \alpha_t \mathbf{1})^\top (p_t - p_{t+1}) - \frac{1}{\eta_t} KL(p_{t+1}, p_t)$$

$$\le \max_{p \in \mathbb{R}_{>0}^N} \left\{ (\ell_t - \alpha_t \mathbf{1})^\top (p_t - p) - \frac{1}{\eta_t} KL(p, p_t) \right\}, \tag{21}$$

where $\mathbf{1} \in \mathbb{R}^N$ represents the all-one vector, and the equality follows from the fact that $\sum_{i=1}^{N} p_{ti} = \sum_{i=1}^{N} p_{t+1,i} = 1$. The maximum in (21) is attained by $p = (p_{ti} \exp(-\eta_t(\ell_{ti} - \alpha_t)))_{i=1}^{N}$. In fact, the objective function is concave in $p$ and its gradient can be expressed as

$$\nabla_p \left( (\ell_t - \alpha_t \mathbf{1})^\top (p_t - p) - \frac{1}{\eta_t} KL(p, p_t) \right) = -(\ell_t - \alpha_t \mathbf{1}) - \frac{1}{\eta_t} \left( (\log p_i)_{i=1}^{N} - (\log p_{ti})_{i=1}^{N} \right),$$

which is the zero vector if and only if $p = (p_{ti} \exp(-\eta_t(\ell_{ti} - \alpha_t)))_{i=1}^{N}$. By substituting this into the objective function, we have

$$\max_{p \in \mathbb{R}_{>0}^d} \left\{ (\ell_t - \alpha_t \mathbf{1})^\top (p_t - p) - \frac{1}{\eta_t} KL(p, p_t) \right\}$$

$$= \sum_{i=1}^{N} \left( (\ell_{ti} - \alpha_t) p_{ti} - \frac{1}{\eta_t} (p_{ti} - p_{ti} \exp(-\eta_t(\ell_{ti} - \alpha_t))) \right)$$

$$= \frac{1}{\eta_t} \sum_{i=1}^{N} p_{ti} \left( \exp(-\eta_t(\ell_{ti} - \alpha_t)) \right) + \eta_t(\ell_{ti} - \alpha_t) - 1) = \frac{1}{\eta_t} \sum_{i=1}^{N} g\left( -\eta_t(\ell_{ti} - \alpha_t) \right).$$

Combining this with (20) and (21), and from $H(p_1) = H(\mathbf{1}/N) = \log N$, we obtain (5). $\qquad \square$

## A.2 Proof of (11)

To show (11), we use the following upper bound on $H(p)$:

**Lemma 4.** *For any $p \in [0, 1]^N$ such that $\|p\|_1 = 1$ and $i^* \in [N]$, we have*

$$H(p) \le (1 - p_i^*) \left( 1 + \log \frac{N - 1}{1 - p_{i^*}} \right). \tag{22}$$

*Proof.* The value of $H(p)$ can be expressed as

$$H(p) = p_{i^*} \log \frac{1}{p_{i^*}} + \sum_{i \in [N] \setminus \{i^*\}} p_i \log \frac{1}{p_i}. \tag{23}$$

The first term can be bounded as

$$p_{i^*} \log \frac{1}{p_{i^*}} = p_{i^*} \log \left(1 + \frac{1 - p_{i^*}}{p_{i^*}}\right) \le p_{i^*} \left(\frac{1 - p_{i^*}}{p_{i^*}}\right) = 1 - p_{i^*}, \tag{24}$$

where the inequality follows from $\log(1 + x) \le x$ that holds for any $x > -1$. When $p_{i^*}$ is fixed, the second term of the right-hand side of (23) is maximized by setting $p_i = \frac{1 - p_{i^*}}{N - 1}$ for all $i \in [N] \setminus \{i^*\}$, and hence, its value can be bounded as

$$\sum_{i \neq [N] \setminus \{i^*\}} p_i \log \frac{1}{p_i} \le (N - 1) \frac{1 - p_{i^*}}{N - 1} \log \frac{N - 1}{1 - p_{i^*}} = (1 - p_{i^*}) \log \frac{N - 1}{1 - p_{i^*}}. \tag{25}$$

Combining this with (23) and (24), we obtain (22). $\qquad\square$

*Proof of* (11). Set $T' = \lfloor 8 \log N \rfloor$. We then have $\eta_t \le 1$ for $t > T'$. From Lemma 1 with $\alpha_t = 0$ for $t \le T'$ and $\alpha_t = \ell_{ti^*}$ for $t > T$, we have

$$R_{Ti^*} \le \frac{\log N}{\eta_1} + \sum_{t=1}^{T'} \frac{1}{\eta_t} \sum_{i=1}^{N} p_{ti} g(\eta_t(-\ell_{ti})) + \sum_{t=T'+1}^{T} \frac{1}{\eta_t} \sum_{i=1}^{N} p_{ti} g(\eta_t(\ell_{ti^*} - \ell_{ti})) + \sum_{t=1}^{T} \left(\frac{1}{\eta_{t+1}} - \frac{1}{\eta_t}\right) H(p_{t+1})$$

$$\le \frac{\log N}{\eta_1} + \sum_{t=1}^{T'} \sum_{i=1}^{N} p_{ti} + (e - 2) \sum_{t=T'+1}^{T} \eta_t \sum_{i=1}^{N} p_{ti}(\ell_{ti} - \ell_{ti^*})^2 + \sum_{t=1}^{T} \left(\frac{1}{\eta_{t+1}} - \frac{1}{\eta_t}\right) H(p_{t+1})$$

$$\le \frac{\log N}{\eta_1} + T' + (e - 2) \sum_{t=T'+1}^{T} \eta_t(1 - p_{ti^*})$$

$$+ \sum_{t=1}^{T-1} \left(\frac{1}{\eta_{t+1}} - \frac{1}{\eta_t}\right) (1 - p_{t+1,i^*}) \left(1 + \log \frac{N - 1}{1 - p_{t+1,i^*}}\right) + \left(\frac{1}{\eta_{T+1}} - \frac{1}{\eta_T}\right) H(p_{T+1})$$

$$\le \sqrt{\frac{\log N}{8}} + 8 \log N + \sqrt{8 \log N}(e - 2) \sum_{t=T'+1}^{T} \frac{1 - p_{ti^*}}{\sqrt{t}}$$

$$+ \frac{1}{2\sqrt{8 \log N}} \sum_{t=2}^{T} \frac{1 - p_{ti^*}}{\sqrt{t - 1}} \left(1 + \log \frac{N - 1}{1 - p_{ti^*}}\right) + \frac{\sqrt{\log N}}{2\sqrt{8T}}$$

$$\le 9 \log N + \frac{1}{4\sqrt{\log N}} \sum_{t=1}^{T} \frac{1 - p_{ti^*}}{\sqrt{t}} \left(1 + \log(N - 1) + \log \frac{1}{1 - p_{ti^*}} + 8\sqrt{2}(e - 2) \log N\right)$$

$$\le 9 \log N + \frac{1}{4\sqrt{\log N}} \sum_{t=1}^{T} \frac{1 - p_{ti^*}}{\sqrt{t}} \left(12 \log N + \log \frac{1}{1 - p_{ti^*}}\right),$$

where the second inequality follows from $g(-y) \le y$ for $y \ge 0$, $g(-y) \le (e - 2)y^2$ for $y \ge -1$, and $\ell_{ti} \in [0, 1]$. The third inequality follows from (22) and the forth inequality follows from $\eta_t = \sqrt{\frac{8 \log N}{t}}$. $\qquad\square$

### A.3 Proof of (12)

We use the following lemma to bound the left-hand side of (12).

**Lemma 5.** *For any $c \in \mathbb{R}$, we have*

$$\max_{x \in (0,1]} x(c - \log x) \le \begin{cases} \exp(c - 1) & c \le 1 \\ c & c > 1 \end{cases}. \tag{26}$$

*Proof.* Define $f : \mathbb{R}_{>0} \to \mathbb{R}$ by $f(x) = x(c - \log x)$. The derivative of $f$ can be expressed as

$$f'(x) = (c - \log x) - 1, \tag{27}$$

which implies that $x = \exp(c - 1)$ is the unique stationary point of $f$. As $f(x)$ is concave in $x \in \mathbb{R}_{>0}$, subject to the constraint of $x \in (0, 1]$, $f(x)$ will be maximized by $x = \min\{1, \exp(c - 1)\}$. By substituting this into $f(x) = x(c - \log x)$, we obtain (26). $\qquad\square$

*Proof of* (12). Set $T' = \lceil (\frac{a}{b})^2 \rceil$. From (26), we have

$$\sum_{t=1}^{T} \frac{x_t}{\sqrt{t}} \left( a - b\sqrt{t} - \log x_t \right) = \sum_{t=1}^{T'} \frac{x_t}{\sqrt{t}} \left( a - b\sqrt{t} - \log x_t \right) + \sum_{t=T'+1}^{T} \frac{x_t}{\sqrt{t}} \left( a - b\sqrt{t} - \log x_t \right)$$

$$\leq \sum_{t=1}^{T'} \frac{a - b\sqrt{t}}{\sqrt{t}} + \sum_{t=T'+1}^{\infty} \frac{1}{\sqrt{t}} \exp\left( a - b\sqrt{t} - 1 \right) \leq a \sum_{t=1}^{T'} \frac{1}{\sqrt{t}} + \exp(a - 1) \sum_{t=T'+1}^{\infty} \frac{\exp\left( -b\sqrt{t} \right)}{\sqrt{t}}.$$

$$(28)$$

The first term can be bounded by

$$\sum_{t=1}^{T'} \frac{1}{\sqrt{t}} \leq 2 \sum_{t=1}^{T'} \frac{1}{\sqrt{t} + \sqrt{t-1}} = 2 \sum_{t=1}^{T'} \left( \sqrt{t} - \sqrt{t-1} \right) = 2\sqrt{T'} \leq 2\sqrt{\left( \frac{a}{b} \right)^2 + 1} \leq \frac{2a}{b} + \frac{b}{a}. \tag{29}$$

Further, as $\exp(-b\sqrt{y})$ is convex in $y \geq 0$, and as its derivative in $y$ can be expressed as $-\frac{b}{2\sqrt{y}} \exp(-b\sqrt{y})$, we have

$$\exp(-b\sqrt{t-1}) - \exp(-b\sqrt{t}) \geq \frac{b}{2\sqrt{t}} \exp(-b\sqrt{t}). \tag{30}$$

From this, we have

$$\exp(a-1) \sum_{t=T'+1}^{\infty} \frac{\exp\left( -b\sqrt{t} \right)}{\sqrt{t}} \leq \frac{2\exp(a-1)}{b} \sum_{t=T'+1}^{\infty} \left( \exp(-b\sqrt{t-1}) - \exp(-b\sqrt{t}) \right)$$

$$= \frac{2}{b} \exp(-b\sqrt{T'} + a - 1) \leq \frac{2}{b} \exp(-1) \leq \frac{1}{b},$$

where the first inequality follows from (30) and the second inequality follows from $T' \geq (\frac{a}{b})^2$. Combining this with (28) and (29), we obtain (12). $\qquad \square$

### A.4 Proof of Theorem 6

We show lower bounds for the following four cases: (i) If $T \leq \frac{N}{\Delta^2}$, $\bar{R}_{Ti^*} = \Omega(\sqrt{TN})$. (ii) If $\frac{C}{\Delta} \leq \frac{N}{\Delta^2} \leq T$, $\bar{R}_{Ti^*} = \Omega(\frac{N}{\Delta})$. (iii) If $\frac{N}{\Delta^2} \leq \frac{C}{\Delta} \leq T$, $\bar{R}_{Ti^*} = \Omega(\sqrt{\frac{CN}{\Delta}})$. (iv) If $\frac{N}{\Delta^2} \leq T \leq \frac{C}{\Delta}$, $\bar{R}_{Ti^*} = \Omega(\sqrt{TN})$. Combining all four cases of (i)–(iv), we obtain (18).

(i) Suppose $T < \frac{N}{\Delta^2}$. Set $\Delta' = \sqrt{\frac{N}{T}}$. We then have $T = \frac{N}{\Delta'^2}$ and $\Delta < \Delta' \leq 1/4$. If $\ell_t \sim \mathcal{D}_{\Delta',i^*}$ for all $t \in [T]$, the environment is then in an adversarial regime with a $(i^*, \Delta, N, C, T)$ self-bounding constraint for any $C \geq 0$, and the regret is bounded as $\bar{R}_{Ti^*} \geq \frac{N}{32\Delta'} = \Omega(\sqrt{TN})$ from Lemma 3.

(ii) Suppose $\frac{C}{\Delta} \leq \frac{N}{\Delta^2} \leq T$. If $\ell_t \sim \mathcal{D}_{\Delta,i^*}$ for all $t \in [T]$, the regret is bounded as $\bar{R}_{Ti^*} \geq \frac{N}{32\Delta}$ for some $i^*$ from Lemma 3. The environment is then in an adversarial regime with a $(i^*, \Delta, N, C, T)$ self-bounding constraint for any $C \geq 0$.

(iii) Suppose $\frac{N}{\Delta^2} \leq \frac{C}{\Delta} \leq T$. Define $\Delta' = \sqrt{\frac{\Delta N}{C}} \leq \Delta$. We then have $\frac{N}{\Delta'} = \sqrt{C\frac{N}{\Delta}}$. Let $T' = \lceil \frac{N}{\Delta'^2} \rceil = \lceil \frac{C}{\Delta} \rceil \leq T$. Consider an environment in which $\ell_t \sim \mathcal{D}_{\Delta',i^*}$ for $t \in [T']$ and $\ell_t \sim \mathcal{D}_{\Delta,i^*}$ for $t \in [T'+1, T]$. From Lemma 3, there then exists $i^* \in [N]$ such that $\bar{R}_{Ti^*} \geq \bar{R}_{T'i^*} \geq \frac{N}{32\Delta'} = \Omega(\sqrt{\frac{CN}{\Delta}})$. Further, we can show that the environment is in an adversarial regime with a $(i^*, \Delta, N, C, T)$ self-bounding constraint. In fact, we have $T'(\Delta - \Delta') \leq \frac{C}{\Delta}(\Delta - \Delta') \leq C$.

(iv) Suppose $\frac{N}{\Delta^2} \leq T \leq \frac{C}{\Delta}$. Set $\Delta' = \sqrt{\frac{N}{T}}$ and consider $\ell_t \sim \mathcal{D}_{\Delta',i^*}$ for all $t \in [T]$. The regret is then bounded as $\bar{R}_{Ti^*} \geq \frac{N}{32\Delta'} = \Omega(\sqrt{TN})$ for some $i^*$, from Lemma 3. We can confirm that the environment is in an adversarial regime with a $(i^*, \Delta, N, C, T)$ self-bounding constraint, as we have $\Delta' T \leq \Delta T \leq C$, where the first and second inequalities follow from $\frac{N}{\Delta^2} \leq T$ and $T \leq \frac{C}{\Delta}$, respectively. $\qquad \square$