# OpenReview forum: "On Optimal Robustness to Adversarial Corruption in Online Decision Problems"
_NeurIPS.cc/2021/Conference — NeurIPS 2021 Poster_

### Official Review · Reviewer_8E6J · 2021-07-12

**Rating:** 7
**Confidence:** 4

**Summary:**

This paper study the expert problem and multi-armed bandit problem in the adversarial regive with self-bounding constraints. Compared to prior work, this paper provides tight lower-bound for both problems and refined upper-bound for the expert setting, achieving matching upper and lower bounds in both settings up to constant and logT factors respectively.

**Main Review:**

This paper nearly closes the theoretical gap between the upper and lower bounds in both problems mentioned above, effectively solved these two problems to completion. The presentation is clear and the analysis is intuitive. It is obviously above the acceptance threshold of the conference.

**Time Spent Reviewing:**

1

---

> ### Author Response · Authors · 2021-08-10
> **Response to Reviewer 8E6J**
>
> Thank you for reading and reviewing this paper, and thank you for the positive feedback.

---

### Official Review · Reviewer_Kpdi · 2021-07-13

**Rating:** 6
**Confidence:** 4

**Summary:**

The paper investigates on the regret bounds for the expert advice problem and the multi-armed bandit in a quite general intermediate regime so called adversarial regime with a self-bounding constraint provided by Zimmert and Seldin (2021) (henceforth ZS21). For the expert advice problem in this general regime, the authors provide two regret bounds and a matching lower bound for an instance of Decreasing Hedge algorithm and algorithms with the second-order bounds. For the multi-armed bandit problem they achieve a lower bound in the mentioned intermediate regime that matches with the state-of-the-art result by ZS21 up to a logarithmic factor.

**Limitations And Societal Impact:**

Limitations:
- As I mentioned earlier, the analysis of the paper follows the ZS21 approach. The main drawback of ZS21 in analyzing Tsallis-INF algorithm was that their analysis technique relies on the uniqueness zero entry (i*) of the self-bounding constraint. One of the important open problem followed by ZS21 is to remove this assumption. This paper also implicitly have the same assumption in their analysis so the authors must clearly state this issue in the paper.

- Two notions of regret have been expressed in the paper: 1- Regret on original version of losses, 2- Regret on corrupted version of losses. It is clear that the regret bounds with these two notions are not comparable but in Table 1 the regret bound by Amir et al., (2020) is compared to the results of Theorems 3 and 4. At least the notions of regret used for evaluation must be represented clearly in the table.

- In MAB problem the authors compare their provided lower bound to ZS21 result, however, recently ZS21 result was improved by a paper titled: "Improved Analysis of Robustness of the Tsallis-INF Algorithm to Adversarial Corruptions in Stochastic Multiarmed Bandits" (Masoudian and Seldin, 2021).  So it is better that the authors compare their lower bound with the improved result.



**Main Review:**

Originality:
- Theorem 3 and Theorem 4 are new bounds for already existing algorithms for the expert advice problem.
The analysis of these two theorems consist of two key steps:  1) Providing upper bound for the regret in terms of probabilities played by the player. 2) Combining this upper bound with the self-bounding constraint.
Lemma 1 in the paper provides the first step for the proof of Theorem 3  that I believe this is almost same bound as Lemma 6 of Amir et al., (2020). Also Theorem 2 in the paper (Theorem 5 of Cesa-Bianchi et al., (2007)) provides the first step for the proof of Theorem 4. The second step, which I think is the most important step, follows the same idea and steps provided by ZS21 for the analysis of Tsallis-INF algorithm for MAB problem that combines self-bounding constraint to their provided upper bound. Hence, I think the novelty is questionable here since the main contribution of these Theorems is applying ZS21 idea to combine the existing upper bounds and self-bounding constraint.
 - Theorem 5 and Theorem 6 provide lower bounds for the regret in expert advice and MAB problem. In spite of the fact that the example construction provided for lower bound is similar to the prior works but the inference steps for the intermediate regimes seems interesting.

Quality:
- There are many minor issues and small bugs in the proofs but after reading the details of the proofs, finally found out they are technically sound.  I noticed small bugs in the analysis steps of the regret upper bounds, Mostly in Theorem 3, but they won't change the order of the regret bounds however they still affect the constants factors in the regret bounds. Generally I suggest that the authors revise the equations at the end of page 6 and page 7 and also construction of example for lower bound in page 8.

Clarity:
- The writing is generally clear. However reading the proofs takes too much energy from a reader since in many equations the intermediate steps are missing.
- There are some small typos and grammatical issues throughout the paper.

Significance:
- As it is mentioned in the paper there is a work by Amir et al., (2020) that studies the expert advice problem with adversary corruptions and give linear regret on the _original version of losses_ w.r.t. total amount of corruption. However, there is no regret bound on the _corrupted version of losses_ and this paper gives a the tight one for this. So the result on expert advice problem seems significant to me but I would still like to see some motivating examples and practical real world examples of such problems.
- The multi-armed bandit part of paper has no important significance since the provided lower bound have quite naive construction and doesn't match completely with the state-of-the-art result by ZS21.


**Time Spent Reviewing:**

16

---

> ### Author Response · Authors · 2021-08-10
> **Response to Reviewer Kpdi**
>
> Thank you for lots of valuable comments.
> We hope the following response addresses your concerns.
>
> > Hence, I think the novelty is questionable here since the main contribution of these Theorems is applying ZS21 idea to combine the existing upper bounds and self-bounding constraint.
>
> Applying the self-bounding technique to hedge algorithms, especially to the decreasing Hedge, required non-trivial considerations.
> For example,
> one technical novelty of ours can be found in Eq.(12),
> which is used to bound the regret for any $\lambda > 0$.
> This inequality has not been provided in existing studies including [ZS21] and [Amir et al., 2021].
>
> > However, there is no regret bound on the corrupted version of losses and this paper gives a the tight one for this. So the result on expert advice problem seems significant to me but I would still like to see some motivating examples and practical real world examples of such problems.
>
> The motivation for considering the regret for the corrupted version of losses is mentioned in Remark 1 (160--164).
> For example, if each expert’s prediction is itself corrupted, the
> player’s performance should be evaluated in terms of the regret on corrupted version of losses,
> as the consequential prediction performance is determined by the losses after corruptions.
> In contrast, if only the observation of the player is corrupted, the performance should be evaluated in terms of
> the regret on original version of losses.
>
> > The multi-armed bandit part of paper has no important significance since the provided lower bound have quite naive construction and doesn't match completely with the state-of-the-art result by ZS21.
>
> From the results of the paper you shared (Masoudian and Seldin, 2021), the gap between the upper and lower bounds seems to be somewhat smaller than in the current manuscript.
> We consider that characterizing the tight bounds (up to constant factors) for MAB is an important future research.
> We will mention this open question in the revised manuscript.
>
>
> > As I mentioned earlier, the analysis of the paper follows the ZS21 approach. The main drawback of ZS21 in analyzing Tsallis-INF algorithm was that their analysis technique relies on the uniqueness zero entry (i*) of the self-bounding constraint. One of the important open problem followed by ZS21 is to remove this assumption. This paper also implicitly have the same assumption in their analysis so the authors must clearly state this issue in the paper.
>
> As you pointed out, this limitation should have been clearly stated.
> The revised version will explicitly note this assumption.
>
> > Two notions of regret have been expressed in the paper: 1- Regret on original version of losses, 2- Regret on corrupted version of losses. It is clear that the regret bounds with these two notions are not comparable but in Table 1 the regret bound by Amir et al., (2020) is compared to the results of Theorems 3 and 4. At least the notions of regret used for evaluation must be represented clearly in the table.
>
> As the reviewer suggests,
> we will specify the different notions of regret in Table 1 in the revised version.
> However,
> we would like to mention that these two notions are not entirely incomparable as stated in Remark 1 (lines 165--170).
> As the difference between 1- Regret and 2- Regret are at most $O(C)$,
> a bound for 2- Regret of $O(\frac{\log N}{\Delta} + C)$ implies a bound for 1- Regret of $O(\frac{\log N}{\Delta} + C)$ as well.
> This is the reason why we have presented the regret bound by Amir et al., (2020) and our bounds in a single table in the current manuscript.
>
> > In MAB problem the authors compare their provided lower bound to ZS21 result, however, recently ZS21 result was improved by a paper titled: "Improved Analysis of Robustness of the Tsallis-INF Algorithm to Adversarial Corruptions in Stochastic Multiarmed Bandits" (Masoudian and Seldin, 2021). So it is better that the authors compare their lower bound with the improved result.
>
> Thank you for sharing this literature information.
> We will definitely add the results of this paper in the revised manuscript.
> We believe that this related work is important as it reduces the gap between the upper and lower bounds for the multi-armed bandit problem.
>
>
> > Generally I suggest that the authors revise the equations at the end of page 6 and page 7 and also construction of example for lower bound in page 8.
> > However reading the proofs takes too much energy from a reader since in many equations the intermediate steps are missing.
> > There are some small typos and grammatical issues throughout the paper.
>
> Thanks for these comments.
> We looked back at the points the reviewer pointed out and found several issues.
> We will revise the manuscript to improve readability.

---

> ### Comment · Reviewer_Kpdi · 2021-08-24
> **Lower Bounds**
>
> Dear Authors,
>
> I have one question and one issue in the proof of Theorem 5.
>
> The proof of Theorem 5 seems sound but there is one minor issue in line 267 case (iv): Seems that you wrote a wrong lower bound for this case. More precisely you must write \sqrt{T \log N} rather than \sqrt{C \logN / \Delta}.
>
> Furthermore, I failed to find the Lemma 2 in Degenne and Perchet (2016) so please either state the proof of this lemma or provide a precise address and statement.

---

> > ### Author Response · Authors · 2021-08-25
> > **Responses to concerns related to the lower bounds**
> >
> > Thank you for sharing your concerns and for the opportunity to respond.
> >
> > > The proof of Theorem 5 seems sound but there is one minor issue in line 267 case (iv): Seems that you wrote a wrong lower bound for this case.
> >
> > Yes.
> > For the case (iv), $\Omega(\sqrt{T \log N})$ is the correct one.
> > We shall fix this in the revision.
> > Thank you for pointing out.
> > (We are reading your point as being about line 261.)
> >
> >
> > > Furthermore, I failed to find the Lemma 2 in Degenne and Perchet (2016) so please either state the proof of this lemma or provide a precise address and statement.
> >
> > For Lemma 2 in our paper,
> > we should have cited Proposition 4 of [Mourtada and Gaïffas (2019)] (the journal version, not the arxiv version).
> > The revised version will include an explicit pointer to this proposition.
> > In addition, we will replace $\epsilon$ with $2 \Delta$ in lines 253-254, in the revised version.
> > We thank you for reading carefully and finding this issue.
> > Also, we apologize for the extra time and effort required to verify the proofs.
> >
> > [Mourtada and Gaïffas (2019)]: J. Mourtada and S. Gaïffas, On the optimality of the Hedge algorithm in the stochastic regime. Journal of Machine Learning Research, 20:1–28, 2019.

---

### Official Review · Reviewer_M84b · 2021-07-16

**Rating:** 6
**Confidence:** 4

**Summary:**

This paper considers the online decision problems with corruptions and provide upper bounds on the regret of the Hedge algorithm and lower bounds for any algorithm.

**Limitations And Societal Impact:**

More discussions on the results may make the big picture and the focus of this paper clearer.

**Main Review:**

This paper presents quite a rich amount of contents, but there seems to be lack of clear discussions. My major concerns are as follows:
1. In the abstract, it states that this work will study the problem of prediction with expert advice and the multi-armed bandit problem. However, somehow I failed to find the results of the problem of prediction with expert advice.
1. Definition 1 introduces the self-bounding constraint. Though this is implied by Zimmert and Seldin [2021], I think that some discussion on the motivation and the interpretation of this definition would be appreciated.
1. This work reviews the existing literature on corruptions, best-of-both-work algorithms and so on. However, I failed to find what is the focus of this work.
1. There are a bunch of theoretical results in this work, but somehow I cannot easily tell what is the specific problem (stochastic/adversarial? with or without corruption?) the result in each section is for. I think more discussions would be appreciated.
1. Furthermore, some experiments to show the regret of the algorithm (maybe the Hedge?) grows as shown in the upper/lower bounds would make the results more convincing.
1. Among the many results, is there any analytic challenge?

=====================================

Thanks you for your response.

Most of my concerns are solved. However, as claimed in the feedback, the results hold for any instances satisfy Definition 1. This suggests that some existing results should be comparable to your work. May you include some comparisons and discussions among the theoretical results in the paper? Numerical comparison is also appreciable but I understand that can be time-consuming.

Anyway, I would like to increase my rating to 6 for the time being.



**Time Spent Reviewing:**

1

---

> ### Author Response · Authors · 2021-08-10
> **Response to Reviewer M84b**
>
> Thank you for all the useful comments.
> We hope the following response addresses your concerns.
>
> > 1. In the abstract, it states that this work will study the problem of prediction with expert advice and the multi-armed bandit problem. However, somehow I failed to find the results of the problem of prediction with expert advice.
>
> Theorems 3, 4 and 5 are results for the problem of prediction with expert advice,
> which is called the *expert problem* in the paper,
> as stated in the first paragraph of the introduction.
> The results for the expert problem are summarized in Table 1 as well.
>
> > 2. Definition 1 introduces the self-bounding constraint. Though this is implied by Zimmert and Seldin [2021], I think that some discussion on the motivation and the interpretation of this definition would be appreciated.
>
> The motivation and the interpretation are discussed right after Definition 1 (lines 144--156).
> As stated in lines 144-156,
> the environments given in Definition 1 includes adversarial settings (lines 145--146),
> stochastic settings (Example 1),
> and stochastic settings with adversarial corruptions (Example 2).
> The motivation for introducing Definition 1 is to provide a comprehensive analysis of these different settings.
>
> > 3. This work reviews the existing literature on corruptions, best-of-both-work algorithms and so on. However, I failed to find what is the focus of this work.
>
> As the title suggests,
> the main focus of this paper is the stochastic regime with adversarial corruptions.
> In order to analyze this regime,
> this paper considers a more comprehensive setting given in Definition 1.
>
>
> > 4. There are a bunch of theoretical results in this work, but somehow I cannot easily tell what is the specific problem (stochastic/adversarial? with or without corruption?) the result in each section is for. I think more discussions would be appreciated.
>
> The theoretical results in this paper can be applied to all of adversarial settings and stochastic setting with/without corruptions as the environments given in Definition 1 includes them as special cases.
> In the revised version,
> we will mention this fact more explicitly.
>
> > 5. Furthermore, some experiments to show the regret of the algorithm (maybe the Hedge?) grows as shown in the upper/lower bounds would make the results more convincing.
>
> Since the main results of this paper are related to theory, the experimental results have been omitted.
> In the evaluation of robustness to adversarial corruptions,
> we believe that theoretical analysis is more convincing than numerical experiments
> as it is nearly impossible to assess robustness under all possible corruptions in experiments.
>
> > 6. Among the many results, is there any analytic challenge?
>
> One analytic challenge in Theorem 3 is to use Eq. (12).
> In the previous work by Amir et al. (2020),
> similar inequalities do not appear though they analyze the same algorithm.

---

> > ### Comment · Reviewer_M84b · 2021-09-01
> > **Reply to author**
> >
> > Thanks you for your response.
> >
> > Most of my concerns are solved. However, as claimed in your feedback, the results hold for any instances satisfy Definition 1. This suggests that some existing results should be comparable to your work. May you include some comparisons and discussions among the theoretical results in the paper? Numerical comparison is also appreciable but I understand that can be time-consuming…
> >
> > Anyway, I would like to increase my rating to 6 for the time being.

---

> > > ### Author Response · Authors · 2021-09-01
> > > **On comparison with existing studies**
> > >
> > > Dear Reviewer M84b,
> > >
> > > Thank you for reading and commenting on our response.
> > >
> > > > This suggests that some existing results should be comparable to your work. May you include some comparisons and discussions among the theoretical results in the paper?
> > >
> > > Yes, there are several comparable existing studies, as you pointed out.
> > > The most relevant of these is the study by Amir et al. (2020), and the comparison with our study is detailed in Table 1, Remark 1 (lines 165--170), and in the response to Reviewer Kpdi.
> > > Briefly speaking, their paper is about the latest research in the stochastic setting with adversarial corruptions, and our results include theirs (as our upper bound implies theirs) and are stronger than theirs.
> > > In other settings (in each of the stochastic and adversarial settings), our bounds are tight up to constant factors, and thus include the results of many existing studies.
> > >
> > > On the other hand, as Reviewer Kpdi and the Area Chair pointed out, our bound is slightly more restrictive than the recent paper by Ito (2021) in that ours relies on the assumption of the uniqueness zero entry of the self-bounding constraint,
> > > though Ito (2021) focuses only on the multi-armed bandit problem.
> > > In the revised version, we will add such weaknesses as well.
> > >
> > >
> > > [Amir et al. (2020)]: I. Amir, I. Attias, T. Koren, Y. Mansour, and R. Livni. Prediction with corrupted expert advice. Advances in Neural Information Processing Systems, 33, 2020.
> > >
> > > [Ito (2021)]: ITO, Shinji. Parameter-Free Multi-Armed Bandit Algorithms with Hybrid Data-Dependent Regret Bounds. In: Conference on Learning Theory. PMLR, 2021. p. 2552-2583.

---

### Official Review · Reviewer_sGi4 · 2021-07-21

**Rating:** 7
**Confidence:** 3

**Summary:**

The paper considered expert problem and multi-armed bandits in stochastic environment with adversarial corruption. In this setting, an adversary modifies stochastic feedback up to total corruption level $C$. The authors analyzed Hedge algorithm with decreasing learning rate and algorithms with second-order regret bounds, and showed that the regret has a $\sqrt{C}$ dependency to the corruption level. Matching lower bound in the expert problem and near-tight lower bound up to logarithmic factor in the multi-armed bandit problem, suggesting the analyzed algorithms achieve optimal robustness.


**Limitations And Societal Impact:**

The authors discussed the limitations and potential social impact of the paper.


**Main Review:**

Strength:

The paper studied an important problem: stochastic online decision making with adversarial corruption. Many recent works focused on this line of research. The key contribution is the refined regret analysis of Hedge algorithms for expert problem and multi-armed bandit problem under corruption. It is interesting to see that in both settings, the regret has a $\sqrt{C}$ dependency to the corruption level. Another contribution is the lower bound of the two problems, showing the analysis is (near) optimal.

One thing to note is that the regret definition in this paper is based on losses after corruption. The authors explained the difference from defining regret based on losses before corruption and analyzed the regret difference in Remark 1.

Concern:

I do not have significant concerns about the paper. One comment is that the authors may want to discuss the technical novelty in Theorem 3 comparing with Amir et al. 2020, since both papers analyzed Hedge algorithm with $\eta_t = O(\sqrt{\log N / t}) learning rate. The authors already provided a comparison on the regret bounds in Remark 1; it would be better to provide more explanation on how the stronger result is achieved.

Quality and clarity:

The writing is clear. The related works are thoroughly discussed. The theoretical analysis looks sound to me although I did not check every step of proof.

References:
I. Amir, I. Attias, T. Koren, Y. Mansour, and R. Livni. Prediction with corrupted expert advice. Advances in Neural Information Processing Systems, 33, 2020.


**Time Spent Reviewing:**

4

---

> ### Author Response · Authors · 2021-08-10
> **Response to Reviewer sGi4**
>
> Thank you for all the useful comments.
> We hope the following response addresses your concerns.
>
> > One comment is that the authors may want to discuss the technical novelty in Theorem 3 comparing with Amir et al. 2020, since both papers analyzed Hedge algorithm with $\eta_t = O(\sqrt{\log N / t})$ learning rate.
>
> Given the analysis by Amir et al. 2020,
> one technical novelty of ours can be found in line 213 in the manuscript.
> Our analysis is given for any value of the parameter $\lambda > 0$
> while the analysis by Amir et al. 2020 can be regarded as
> that for a special case in which $\lambda = \Theta (1)$.
> Another novelty is to use Eq.(12),
> which has not been provided in the previous work.
>
> > it would be better to provide more explanation on how the stronger result is achieved.
>
> Thanks for this comment.
> An overview of the methods used to obtain the stronger result is given Lines 78-86.
> In the revised version,
> we will add more explanation, such as the one we wrote in our reply above.

---

### Decision · Program_Chairs · 2021-09-27

**Decision:**

Accept (Poster)

**Comment:**

The reviewers unanimously support acceptance of the paper.